# Event-Triggered Optimal Tracking Control for Uncertain Nonlinear System Based on Reinforcement Learning

Yuanhao Wang
*Navigation College*
*Dalian Maritime University*
Dalian, China
wangyuanhao2024@163.com

Weiwei Bai
*Navigation College*
*Dalian Maritime University*
Dalian, China
baiweiwei_dl@163.com

*Abstract*—In this paper, an event-triggered optimal tracking control problem is studied for uncertain nonlinear systems based on reinforcement learning (RL). Firstly, a class of nonlinear dynamic systems with general uncertainty is considered and the augmented system comprising tracking error and reference signal is constructed. Secondly, an improved adaptive dynamic programming (ADP) technique, involving actor-critic algorithm and fuzzy logic systems, is developed to solve the Hamilton–Jacobi–Bellman (HJB) equation with respect to nominal augmented system. Thirdly, in order to reduce the mechanical wear of actuator and energy consumption, event-triggered mechanism is performed in controller updating. Finally, stability analysis proofs that all signals are uniformly ultimately bounded (UUB) in the closed-loop system via Lyapunov theory. Simulation results verify feasibility of proposed scheme.

*Index Terms*—ADP, event-triggered, reinforcement learning, nonlinear, fuzzy logic systems, tracking control.

## I. INTRODUCTION

Reinforcement Learning (RL) as an effective technique has competent in facilitating adaptive optimization strategy [1], [2]. Generally, optimization is implemented via seeking minimized or maximized cost function to solve the Hamilton–Jacobi–Bellman (HJB) equation [3]. However, there exists a challenge about acquiring analytic solution of HJB equation directly for nonlinear dynamic systems [4]. Therefore many researchers proposed numerical solution of HJB equation [5]. Adaptive dynamic programming (ADP) as an advanced numerical solving method, has been widely applied to achieve the optimal tracking control of nonlinear systems.

In contrast to traditional dynamic programming, ADP can be utilized to design optimal controller forward in time, which effectively avoids "curse of dimensionality" [6], [7]. In addition, an improved ADP framework consists of actor-critic algorithm and fuzzy logic systems is constructed. So far, there have been many scholars devoting to developing ADP techniques [8]–[10]. In [11], ADP method was implemented to solve a new neuro-optimal control problem of nonlinear dynamic systems by employing one critic and two actor networks. In [12], a neural-network-based ADP method was developed to solve the optimal tracking control problem of a class of nonlinear systems with unmatched uncertainties. In [13], linear singularly perturbed system was studied via employing ADP framework to achieve optimal control. These literatures concentrated on application and development of ADP and RL, but they did not consider the condition with mechanical wear of actuator and energy consumption. As a result, it is of necessity to perform event-triggered mechanism in control design for reducing mechanical wear and saving energy in actual engineering practice [14].

The key of event-triggered control algorithm is triggering threshold [14]. When signal exceeds triggering threshold, control policy will be updated [15], [16]. In this paper, an event-triggered optimal tracking control scheme for uncertain nonlinear systems based on RL is developed. There are two main contributions:

(1) An improved ADP and RL algorithm involving actor-critic and fuzzy logic systems is developed, which develops the optimal control strategy and effectively balances the tracking control performance and control costs.

(2) Event-triggered mechanism is performed in controller design, the unnecessary control input is avoided, achieving the reduction of mechanical wear and the energy saving in engineering practice.

The organization of this paper is shown as follows. System dynamic description and fuzzy logic systems are stated in Section II. Optimal controller and event-triggered controller are designed in Sections III and IV, respectively. Stability analysis, simulation and conclusion are shown in Sections V, VI and VII, respectively.

## II. PROBLEM FORMULATION AND PRELIMINARIES

### A. System dynamic description

Consider a class of continuous-time nonlinear dynamic systems which can be described by

$$\dot{x}(t) = f(x(t)) + g(x(t))u(t) + \mathcal{D}(x(t)) \tag{1}$$

where $x(t) \in \mathbb{R}^n$ is the state variable, $u(t) \in \mathbb{R}^m$ is the control input, $f(x(t)) \in \mathbb{R}^n$ and $g(x(t)) \in \mathbb{R}^{n \times m}$ are the unknown smooth function and unknown smooth function

matrix respectively, $\mathcal{D}(x(t))$ is the unknown disturbance with $\|\mathcal{D}(x(t))\| \leq \lambda_{\mathcal{D}}$ and $\lambda_{\mathcal{D}}$ is a positive parameter.

To achieve tracking control, a reference signal is given by

$$\dot{r}(t) = \delta(r(t)) \tag{2}$$

where $r(t) \in \mathbb{R}^n$ is a bounded desired trajectory and $\delta(r(t))$ is a Lipschitz continuous function. Let the tracking error be

$$e(t) = x(t) - r(t) \tag{3}$$

Combining equations (1), (2) and (3), one can yield the following dynamic of tracking error

$$\dot{e}(t) = f(x(t)) + g(x(t))u(t) + \mathcal{D}(x(t)) - \delta(r(t)) \tag{4}$$

Note that $x(t) = e(t) + r(t)$, equation (4) can be rewritten as

$$\begin{aligned}\dot{e}(t) =& f(e(t)+r(t)) - \delta(r(t)) + g(e(t)+r(t))u(t)\\&+ \mathcal{D}(e(t)+r(t))\end{aligned} \tag{5}$$

For the sake of facilitating description, define $\xi(t) = [e^{\mathrm{T}}(t), r^{\mathrm{T}}(t)]^{\mathrm{T}} \in \mathbb{R}^{2n}$, and then dynamic systems (2) and (5) can be augmented as a concise form

$$\dot{\xi}(t) = F(\xi(t)) + G(\xi(t))u(t) + \Delta \mathbb{D}(\xi(t)) \tag{6}$$

where $F(\xi(t))$ and $G(\xi(t))$ are new matrices and $\Delta \mathbb{D}(\xi(t))$ can be still regarded as a new uncertain term. In particular, $F(\xi(t)) = \begin{bmatrix} f(e(t)+r(t)) - \delta(t) \\ \delta(t) \end{bmatrix}$, $G(\xi(t)) = \begin{bmatrix} g(e(t)+r(t)) \\ 0_{n \times m} \end{bmatrix}$ and $\Delta \mathbb{D}(\xi(t)) = \begin{bmatrix} \mathcal{D}(e(t)+r(t)) \\ 0_{n \times 1} \end{bmatrix}$.

Undoubtedly, the new uncertain term $\Delta \mathbb{D}(\xi(t))$ is still upper bounded since

$$\|\Delta \mathbb{D}(\xi(t))\| = \|\mathcal{D}(e(t)+r(t))\| = \|\mathcal{D}(x(t))\| \leq \lambda_{\mathcal{D}} \tag{7}$$

To accomplish tracking control of dynamic system (1) to reference signal (2), the feedback controller $u(\xi)$ will be constructed. One can yield that the closed-loop system is asymptotically stable under the controller $u(\xi)$ for the uncertain and bounded term $\Delta \mathbb{D}(\xi(t))$. Therefore, the optimal control policy can be applied by considering appropriate cost function of the subsequent nominal system the same as that in [5].

*B. Fuzzy logic systems*

Define a nonlinear continuous function $P(x)$ over a compact set $\mathbb{U}$, for any constant $\varepsilon > 0$, there exists fuzzy logic systems $\omega^{\mathrm{T}} \varphi(x)$ such that [17]

$$\sup_{x \in \mathbb{U}} |P(x) - \omega^{\mathrm{T}} \varphi(x)| \leq \varepsilon \tag{8}$$

where $x = [x_1, \ldots, x_j]^{\mathrm{T}}$ is the input vector of fuzzy logic systems, $\omega = [\omega_1, \omega_2, \ldots, \omega_L]^{\mathrm{T}} \in \mathbb{R}^L$ is the degree of membership and $L > 1$ is the number of fuzzy rules, $\varepsilon$ is the fuzzy minimum approximation error. $\varphi(x) =$

$[\varphi_1(x), \varphi_2(x), \ldots, \varphi_L(x)]^{\mathrm{T}}$ is fuzzy basic function vector and $\varphi_i(x)$ is selected as follows:

$$\varphi_i(x) = \frac{\prod_{i=1}^{j} \mu_{F_i^l}(x_i)}{\sum_{i=1}^{N} \left( \prod_{i=1}^{j} \mu_{F_i^l}(x_i) \right)}, (i = 1, \ldots, L) \tag{9}$$

where $F_i^l(i = 1, \ldots, j; l = 1, \ldots, N)$ is the fuzzy set and $\mu_{F_i^l}(x_i)$ is the membership function.

## III. OPTIMAL CONTROL DESIGN

In this section, ADP comprising actor-critic algorithm and fuzzy logic systems will be employed to design the value function $L^*(\xi)$ and control policy $u^*(\xi)$, and design degree of membership update laws.

In actor-critic framework, value function and control policy are approximated by critic and actor fuzzy systems, respectively. Optimal cost function (13) and feedback controller (15) represent value function and control policy for optimal tracking control problem, respectively.

Consider the nominal part of the augmented system (6), that is

$$\dot{\xi}(t) = F(\xi(t)) + G(\xi(t))u(t) \tag{10}$$

For the nominal system (10), this cost function is considered

$$L(\xi) = \int_t^{\infty} Q(\tau) + u(\tau)^{\mathrm{T}} Ru(\tau) d\tau \tag{11}$$

where $Q(\xi) = \xi^{\mathrm{T}} \mathcal{Q} \xi$, $R = R^{\mathrm{T}}$. $\mathcal{Q}$ and $R$ are positive defined matrix.

Subsequently, one can define the Hamiltonian of the optimal problem

$$\begin{aligned}H(\xi, u(\xi)) =& Q(\xi) + u(\xi)^{\mathrm{T}} Ru(\xi) \\&+ \nabla^{\mathrm{T}} L(\xi) [F(\xi) + G(\xi)u(\xi)]\end{aligned} \tag{12}$$

where $\nabla L(\xi)$ represents the partial derivative of $L(\xi)$.

Generally, as long as finding the optimal cost function can we derive the optimal controller. The infinitesimal version of cost function is regarded as the optimal cost function, one has

$$L^*(\xi) = \min \int_t^{\infty} Q(\tau) + u(\tau)^{\mathrm{T}} Ru(\tau) d\tau \tag{13}$$

The optimal cost function is the solution of the HJB equation which satisfies

$$\begin{aligned}H(\xi, u^*(\xi), L^*(\xi)) =& Q(\xi) + u^*(\xi)^{\mathrm{T}} Ru^*(\xi) \\&+ \nabla^{\mathrm{T}} L(\xi) [F(\xi) + G(\xi)u^*(\xi)] = 0\end{aligned} \tag{14}$$

Consequently, the optimal feedback controller is yielded

$$u^*(\xi) = -\frac{1}{2} R^{-1} G^{\mathrm{T}}(\xi) \nabla L^*(\xi) \tag{15}$$

One need to solve the HJB equation (14) and obtain the optimal controller (15) for nominal system (10). However, the solution of HJB equation (14) is difficult to be obtained directly. Therefore, fuzzy logic systems and adaptive actor-critic will be utilized to find its estimated solution.

Fuzzy logic systems are employed to reconstruct the value function $L^*(\xi)$

$$L^*(\xi) = \omega^{\mathrm{T}} \varphi(\xi) + \varepsilon(\xi) \tag{16}$$

where $\omega$ is the degree of membership of fuzzy logic systems, $\varphi(\xi)$ is the fuzzy basis function and $\varepsilon(\xi)$ is the unknown fuzzy approximate error.

Considering (15) and (16) yields the optimal controller described by fuzzy logic systems as

$$u^*(\xi) = -\frac{1}{2} R^{-1} G^{\mathrm{T}}(\xi) \left[ \nabla^{\mathrm{T}} \varphi(\xi) \omega + \nabla \varepsilon(\xi) \right] \tag{17}$$

In order to clearly analyze, define a non-negative matrix

$$A(\xi) = \nabla \varphi(\xi) G(\xi) R^{-1} G(\xi) \nabla^{\mathrm{T}} \varphi(\xi) \tag{18}$$

One can derive the HJB equation reconstructed by fuzzy logic systems, combining with (16), (17) and (18), one has

$$H(\xi, u^*(\xi), L^*(\xi)) = Q(\xi) + \omega^{\mathrm{T}} \nabla \varphi(\xi) F(\xi) \\ - \frac{1}{4} \omega^{\mathrm{T}} A(\xi) \omega + \varepsilon_{HJB} = 0 \tag{19}$$

and the residual error $\varepsilon_{HJB}$ is expressed as

$$\varepsilon_{HJB} = \nabla^{\mathrm{T}} \varepsilon(\xi) \left( F(\xi) + G(\xi) u^*(\xi) \right) \\ + \frac{1}{4} \nabla^{\mathrm{T}} \varepsilon(\xi) G(\xi) R^{-1} G^{\mathrm{T}}(\xi) \nabla \varepsilon(\xi) \\ + \frac{1}{2} \nabla^{\mathrm{T}} \varepsilon(\xi) G(\xi) R^{-1} G^{\mathrm{T}}(\xi) \nabla^{\mathrm{T}} \varphi(\xi) \omega \tag{20}$$

The estimation of value function $L^*(\xi)$ and control policy $u^*(\xi)$ can be constructed by critic and actor fuzzy, respectively.

$$\hat{L}^*(\xi) = \hat{\omega}_c^{\mathrm{T}} \varphi(\xi) \tag{21}$$

$$\hat{u}^*(\xi) = -\frac{1}{2} R^{-1} G^{\mathrm{T}}(\xi) \nabla^{\mathrm{T}} \varphi(\xi) \hat{\omega}_a \tag{22}$$

where $\hat{\omega}_a$ is the actor estimated degree of membership and $\hat{\omega}_c$ is the critic estimated degree of membership.

Noticing (21) and (22), one can derive the following estimated Hamiltonian

$$\hat{H}\left(\xi, \hat{u}^*(\xi), \hat{L}^*(\xi)\right) = Q(\xi) + \frac{1}{4} \hat{\omega}_a^{\mathrm{T}} A(\xi) \hat{\omega}_a \\ + \hat{\omega}_c^{\mathrm{T}} \nabla \varphi(\xi) F(\xi) - \frac{1}{2} \hat{\omega}_c^{\mathrm{T}} A(\xi) \hat{\omega}_a \tag{23}$$

To obtain the degree of membership update laws of fuzzy logic systems, defining the objective function as $E_c = \frac{1}{2} e_c^{\mathrm{T}} e_c$, where $e_c = \hat{H}\left(\xi, \hat{u}^*(\xi), \hat{L}^*(\xi)\right) - H(\xi, u^*(\xi), L^*(\xi))$ is the Bellman error. In order to conquer the difficulties of searching controller and adaptive laws, the following assumption is made and the additional term can be constructed to improve the learning process.

*Assumption 1:* [5] Define $L_s(\xi)$ is a continuous differentiable Lyapunov function candidate satisfying

$$\dot{L}_s(\xi) = \nabla^{\mathrm{T}} L_s(\xi) \left( F(\xi) + u^*(\xi) \right) < 0 \tag{24}$$

and then, there exists a positive matrix $\mathfrak{K} \in \mathbb{R}^{2n \times 2n}$ ensuring that

$$\nabla^{\mathrm{T}} L_s(\xi) \left( F(\xi) + u^*(\xi) \right) = -\nabla^{\mathrm{T}} L_s(\xi) \mathfrak{K} \nabla L_s(\xi) \\ \leq -\lambda_{\min}(\mathfrak{K}) \nabla \| L_s(\xi) \|^2 \tag{25}$$

Based on the gradient decent, degree of membership update laws of fuzzy logic systems are designed, by considering these two Hamilton functions $H(\xi, u^*(\xi), L^*(\xi))$ and $\hat{H}\left(\xi, \hat{u}^*(\xi), \hat{L}^*(\xi)\right)$, one has

$$\dot{\hat{\omega}}_a = -\alpha_a \left( \frac{1}{2} A(\xi) \hat{\omega}_a - \frac{1}{2} A(\xi) \hat{\omega}_c \right) \\ \times \left( Q(\xi) + \frac{1}{4} \hat{\omega}_a^{\mathrm{T}} A(\xi) \hat{\omega}_a + \hat{\omega}_c^{\mathrm{T}} \nabla \varphi(\xi) F(\xi) \right. \\ \left. - \frac{1}{2} \hat{\omega}_c^{\mathrm{T}} A(\xi) \hat{\omega}_a \right) + \frac{1}{2} \alpha_s \nabla \varphi(\xi) G R^{-1} G^{\mathrm{T}} \nabla L_s(\xi) \tag{26}$$

$$\dot{\hat{\omega}}_c = -\alpha_c \left( \nabla \varphi(\xi) F(\xi) - \frac{1}{2} A(\xi) \hat{\omega}_a \right) \\ \times \left( Q(\xi) + \frac{1}{4} \hat{\omega}_a^{\mathrm{T}} A(\xi) \hat{\omega}_a + \hat{\omega}_c^{\mathrm{T}} \nabla \varphi(\xi) F(\xi) \right. \\ \left. - \frac{1}{2} \hat{\omega}_c^{\mathrm{T}} A(\xi) \hat{\omega}_a \right) + \frac{1}{2} \alpha_s \nabla \varphi(\xi) G R^{-1} G^{\mathrm{T}} \nabla L_s(\xi) \tag{27}$$

where $\alpha_a$ and $\alpha_c$ are the basis learning parameters of actor and critic systems, respectively, and $\alpha_s$ is the adjustable parameter for the additional term.

## IV. EVENT-TRIGGERED CONTROL IMPLEMENTATION

The event triggering mechanism is defined as

$$u_e^*(\xi(t)) = u^*(\xi(t_d)), \forall t \in [t_d, \ t_{d+1}) \tag{28}$$

$$t_{d+1} = \inf \left\{ t \in \mathbb{R} | \ |\Gamma(t)| \geq \Delta |u_e^*(\xi(t))| + M \right\}, t_1 = 0 \tag{29}$$

where the event-triggered error $\Gamma(t) = u^*(\xi(t_d)) - u_e^*(\xi(t))$, the controller update time is $t_d$, $d \in Z^+$. Define the proper parameters $0 < \Delta < 1$ and $M > 0$.

When event is not triggered, the control policy will be chosen as $u^*(\xi(t_d))$. Otherwise, control policy will be updated and marked as $u_e^*(\xi(t_{d+1}))$. Assume two continuous and time-varying parameters $\rho_1(t)$ and $\rho_2(t)$, which results in $u^*(\xi(t)) = (1 + \rho_1(t)\Delta) u_e^*(\xi(t)) + \rho_2(t) M$ where $|\rho_1(t)| \leq 1$ and $|\rho_2(t)| \leq 1$. And then, the event-triggered controller can be rewritten as

$$u_e^*(\xi(t)) = \frac{u^*(\xi(t)) - \rho_2(t) M}{1 + \rho_1(t)\Delta} \tag{30}$$

Using (17) and (30) can yield that

$$u_e^*(\xi(t)) = -\frac{1}{2\rho} R^{-1} \left[ G^{\mathrm{T}}(\xi(t)) \nabla^{\mathrm{T}} \varphi(\xi(t)) \omega + \varepsilon_e(\xi(t)) \right] \tag{31}$$

where $\rho = 1 + \rho_1(t)\Delta$, $\varepsilon_e(\xi(t)) = \nabla \varepsilon(\xi(t)) + 2\rho_2(t) R M$.

Similarly, based on critic fuzzy logic systems, the estimated event-triggered controller can be obtained, one has

$$\hat{u}_e^*(\xi(t)) = -\frac{1}{2\rho}R^{-1}G^{\mathrm{T}}(\xi(t))\nabla^{\mathrm{T}}\varphi(\xi(t))\hat{\omega}_a \qquad (32)$$

Considering the HJB equation (14), value function (21) and event-triggered controller (32), one can yield the following Hamilton function as

$$\hat{H}_e\left(\xi(t),\hat{u}_e^*(\xi(t)),\hat{L}^*(\xi(t))\right)$$
$$= Q(\xi(t)) + \frac{1}{4\rho^2}\hat{\omega}_a^{\mathrm{T}}A(\xi(t))\hat{\omega}_a + \hat{\omega}_c^{\mathrm{T}}\nabla\varphi(\xi(t))F(\xi(t))$$
$$- \frac{1}{2\rho}\hat{\omega}_c^{\mathrm{T}}A(\xi(t))\hat{\omega}_a$$
$$(33)$$

Subsequently, degree of membership update laws with respect to event-triggered mechanism can be constructed, one has

$$\dot{\hat{\omega}}_{ae} = -\alpha_a\left(\frac{1}{2\rho^2}A(\xi(t))\hat{\omega}_a - \frac{1}{2\rho}A(\xi(t))\hat{\omega}_c\right)$$
$$\times\left(Q(\xi(t)) + \frac{1}{4\rho^2}\hat{\omega}_a^{\mathrm{T}}A(\xi(t))\hat{\omega}_a\right.$$
$$\left. + \hat{\omega}_c^{\mathrm{T}}\nabla\varphi(\xi(t))F(\xi(t)) - \frac{1}{2\rho}\hat{\omega}_c^{\mathrm{T}}A(\xi)\hat{\omega}_a\right)$$
$$\qquad (34)$$
$$+ \frac{1}{2}\alpha_s\nabla\varphi(\xi(t))GR^{-1}G^{\mathrm{T}}\nabla\ L_s(\xi(t))$$

$$\dot{\hat{\omega}}_{ce} = -\alpha_c\left(\nabla\varphi(\xi(t))F(\xi(t)) - \frac{1}{2\rho}A(\xi(t))\hat{\omega}_a\right)$$
$$\times\left(Q(\xi(t)) + \frac{1}{4\rho^2}\hat{\omega}_a^{\mathrm{T}}A(\xi(t))\hat{\omega}_a\right.$$
$$\left. + \hat{\omega}_c^{\mathrm{T}}\nabla\varphi(\xi(t))F(\xi(t)) - \frac{1}{2\rho}\hat{\omega}_c^{\mathrm{T}}A(\xi(t))\hat{\omega}_a\right)$$
$$\qquad (35)$$
$$+ \frac{1}{2}\alpha_s\nabla\varphi(\xi(t))GR^{-1}G^{\mathrm{T}}\nabla\ L_s(\xi(t))$$

*Theorem 1:* Considering the dynamic system (1), the optimal feedback controller (22), event-triggered controller (32) and the degree of membership update laws (26), (27), (34) and (35) are developed. Based on Lyapunov theory, all signals are uniformly ultimately bounded (UUB) in the closed-loop system.

For the sake of investigating the stability of error dynamics and close-loop states, the following assumption is given by

*Assumption 2:* On a compact set $\Omega$, $G(\xi)$, $\nabla\varphi(\xi)$, $\nabla\varepsilon(\xi)$, $\xi^*$ and $\varepsilon_{HJB}$ are bounded. $\|G(\xi)\| \leq \lambda_g$, $\|\nabla\varphi(\eta)\| \leq \lambda_\varphi$, $\|\nabla\varepsilon(\eta)\| \leq \lambda_\varepsilon$, $\|\xi^*\| \leq \lambda_\xi$ and $\|\varepsilon_{HJB}\| \leq \lambda_{HJB}$, where $\lambda_g$, $\lambda_\varphi$, $\lambda_\varepsilon$, $\lambda_\xi$ and $\lambda_{HJB}$ are positive constants.

## V. STABILITY ANALYSIS

In this section, Lyapunov theory will be employed to demonstrate Theorem 1.

*Case1* : Event are not triggered. Consider the feedback controller (22) and the related degree of membership update laws (26) and (27).

According to HJB equation (19), it can be transformed as

$$Q(\xi) = -\omega^{\mathrm{T}}\nabla\varphi(\xi)F(\eta) + \frac{1}{4}\omega^{\mathrm{T}}A(\xi)\omega - \varepsilon_{HJB} \qquad (36)$$

Considering the degree of membership update laws (26) and (27), combining with $\widetilde{\omega}_a = -\dot{\hat{\omega}}_a$ and $\widetilde{\omega}_c = -\dot{\hat{\omega}}_c$, one has

$$\dot{\hat{\omega}}_a = -\alpha_a\left(-\frac{1}{2}A(\xi)\hat{\omega}_a + \frac{1}{2}A(\xi)\hat{\omega}_c\right)$$
$$\times\left(Q(\xi) + \frac{1}{4}\hat{\omega}_a^{\mathrm{T}}A(\xi)\hat{\omega}_a + \hat{\omega}_c^{\mathrm{T}}\nabla\varphi(\xi)F(\xi)\right.$$
$$\left. - \frac{1}{2}\hat{\omega}_c^{\mathrm{T}}A(\xi)\hat{\omega}_a\right) - \frac{1}{2}\alpha_s\nabla\varphi(\xi)GR^{-1}G^{\mathrm{T}}\nabla\ L_s(\xi)$$
$$(37)$$

$$\dot{\hat{\omega}}_c = -\alpha_c\left(-\nabla\varphi(\xi)F(\eta) + \frac{1}{2}A(\xi)\hat{\omega}_a\right)$$
$$\times\left(Q(\xi) + \frac{1}{4}\hat{\omega}_a^{\mathrm{T}}A(\xi)\hat{\omega}_a + \hat{\omega}_c^{\mathrm{T}}\nabla\varphi(\xi)F(\xi)\right.$$
$$\left. - \frac{1}{2}\hat{\omega}_c^{\mathrm{T}}A(\xi)\hat{\omega}_a\right) - \frac{1}{2}\alpha_s\nabla\varphi(\xi)GR^{-1}G^{\mathrm{T}}\nabla\ L_s(\xi)$$
$$(38)$$

Then the following Lyapunov function can be chosen as

$$S(t) = \frac{1}{2\alpha_a}\widetilde{\omega}_a^{\mathrm{T}}\widetilde{\omega}_a + \frac{1}{2\alpha_c}\widetilde{\omega}_c^{\mathrm{T}}\widetilde{\omega}_c + \frac{\alpha_s}{\alpha_a}L_s(\xi) + \frac{\alpha_s}{\alpha_c}L_s(\xi)$$
$$(39)$$

its derivative is

$$\dot{S}(t) = \frac{1}{\alpha_a}\widetilde{\omega}_a^{\mathrm{T}}\dot{\widetilde{\omega}}_a + \frac{1}{\alpha_c}\widetilde{\omega}_c^{\mathrm{T}}\dot{\widetilde{\omega}}_c + \frac{\alpha_s}{\alpha_a}\nabla^{\mathrm{T}}L_s(\xi)\dot{\xi} + \frac{\alpha_s}{\alpha_c}\nabla^{\mathrm{T}}L_s(\xi)\dot{\xi}$$
$$= \left(\widetilde{\omega}_c^{\mathrm{T}}\nabla\varphi(\xi)F(\xi) - \frac{1}{4}\omega^{\mathrm{T}}A(\xi)\omega - \frac{1}{4}\hat{\omega}_a^{\mathrm{T}}A(\xi)\hat{\omega}_a\right.$$
$$+ \varepsilon_{HJB} + \frac{1}{2}\hat{\omega}_c^{\mathrm{T}}A(\xi)\hat{\omega}_a\right) \times \left(-\widetilde{\omega}_c^{\mathrm{T}}\nabla\varphi(\xi)F(\xi)\right.$$
$$\left. + \frac{1}{2}\widetilde{\omega}_a^{\mathrm{T}}A(\xi)\hat{\omega}_c + \frac{1}{2}\widetilde{\omega}_c^{\mathrm{T}}A(\xi)\hat{\omega}_a - \frac{1}{2}\widetilde{\omega}_a^{\mathrm{T}}A(\xi)\hat{\omega}_a\right)$$
$$- \frac{\alpha_s}{2\alpha_a}\widetilde{\omega}_a^{\mathrm{T}}\nabla\varphi(\xi)GR^{-1}G^{\mathrm{T}}\ \nabla\ L_s(\xi)$$
$$- \frac{\alpha_s}{2\alpha_c}\widetilde{\omega}_c^{\mathrm{T}}\nabla\varphi(\xi)GR^{-1}G^{\mathrm{T}}\ \nabla\ L_s(\xi)$$
$$+ \frac{\alpha_s}{\alpha_a}\nabla^{\mathrm{T}}L_s(\xi)\dot{\xi} + \frac{\alpha_s}{\alpha_c}\nabla^{\mathrm{T}}L_s(\xi)\dot{\xi}$$
$$(40)$$

Substituting (22) into (10) and observing the dynamic system $\dot{\xi}^* = F(\xi) + G(\xi)u^*(\xi)$ with optimal controller $u^*(\xi)$, one can acquire

$$\nabla\varphi(\xi)F(\xi) = \nabla\varphi(\xi)\dot{\xi} + \frac{1}{2}\nabla\varphi(\xi)R^{-1}\nabla^{\mathrm{T}}\varphi(\xi)\hat{\omega}_a \quad (41)$$

$$\dot{\xi} = \dot{\xi}^* + \frac{1}{2}GR^{-1}G^{\mathrm{T}}\left(\nabla^{\mathrm{T}}\varphi(\xi)\widetilde{\omega}_a + \nabla\varepsilon(\xi)\right) \quad (42)$$

Considering above formulations, one can further derive that

$$
\begin{aligned}
\dot S(t) =& \Big( \widetilde\omega_c^{\mathrm T} \nabla\varphi(\xi)\dot\xi^* + \frac{1}{2}\widetilde\omega_c^{\mathrm T}\nabla\varphi(\xi)GR^{-1}G^{\mathrm T}\nabla\varepsilon(\xi) \\
& + \frac{1}{2}\widetilde\omega_c^{\mathrm T} A(\xi)\widetilde\omega_a - \frac{1}{2}\widetilde\omega_a^{\mathrm T} A(\xi)\omega + \frac{1}{4}\widetilde\omega_a^{\mathrm T} A(\xi)\widetilde\omega_a + \varepsilon_{HJB} \Big) \\
& \times \Big( -\widetilde\omega_c^{\mathrm T}\nabla\varphi(\xi)\dot\xi^* - \frac{1}{2}\widetilde\omega_c^{\mathrm T}\nabla\varphi(\xi)GR^{-1}G^{\mathrm T}\nabla\varepsilon(\xi) \\
& - \widetilde\omega_c^{\mathrm T} A(\xi)\widetilde\omega_a - \frac{1}{2}\widetilde\omega_a^{\mathrm T} A(\xi)\widetilde\omega_a \Big) \\
& - \frac{\alpha_s}{2\alpha_a}\widetilde\omega_a^{\mathrm T}\nabla\varphi(\xi)GR^{-1}G^{\mathrm T}\nabla\, L_s(\xi) \\
& - \frac{\alpha_s}{2\alpha_c}\widetilde\omega_c^{\mathrm T}\nabla\varphi(\xi)GR^{-1}G^{\mathrm T}\nabla\, L_s(\xi) \\
& + \frac{\alpha_s}{\alpha_a}\nabla^{\mathrm T} L_s(\xi)\dot\xi + \frac{\alpha_s}{\alpha_c}\nabla^{\mathrm T} L_s(\xi)\dot\xi
\end{aligned}
\tag{43}
$$

Next, equation (43) can be expended to conduct mathematical operations based on Assumption 2 and yields that

$$
\begin{aligned}
\dot S(t) \le & -\lambda_1(\|\widetilde\omega_a\|)^4 - \lambda_2(\|\widetilde\omega_c\|)^2 + \lambda_3 \\
& + \frac{\alpha_s}{2\alpha_a}\nabla^{\mathrm T} L_s(\xi)\; GR^{-1}G^{\mathrm T}\nabla\varepsilon(\xi) \\
& + \frac{\alpha_s}{\alpha_a}\nabla^{\mathrm T} L_s(\xi)(F(\xi)+Gu^*(\xi)) \\
& + \frac{\alpha_s}{2\alpha_c}\nabla^{\mathrm T} L_s(\xi) GR^{-1}G^{\mathrm T}\nabla\varepsilon(\xi) \\
& + \frac{\alpha_s}{\alpha_c}\nabla^{\mathrm T} L_s(\xi)(F(\xi)+Gu^*(\xi))
\end{aligned}
\tag{44}
$$

where $\lambda_1$, $\lambda_2$ and $\lambda_3$ are positive constants.

Considering Assumption 1 and equation (44), one can further derive that

$$
\begin{aligned}
\dot S(t) \le & -\lambda_1(\|\widetilde\omega_a\|)^4 - \lambda_2(\|\widetilde\omega_c\|)^2 + \lambda_\partial \\
& - \lambda_{\min}(\mathfrak{K})\alpha_s(\frac{1}{\alpha_a}+\frac{1}{\alpha_c})(\|\nabla\, L_s(\xi)\| \\
& - \frac{\lambda_g{}^2\lambda_\varepsilon{}^2(\|R^{-1}\|)^2}{4\lambda_{\min}(\mathfrak{K})})^2
\end{aligned}
\tag{45}
$$

where $\lambda_\partial = \lambda_3 + \frac{\lambda_g{}^4\lambda_\varepsilon{}^4(\|R^{-1}\|)^4}{16\lambda_{\min}(\mathfrak{K})}$.

As a result, if $\|\widetilde\omega_a\| \ge \sqrt[4]{\frac{\lambda_\partial}{\lambda_1}}$ or $\|\widetilde\omega_c\| \ge \sqrt{\frac{\lambda_\partial}{\lambda_2}}$ or $\|\nabla\, L_s(\xi)\| \ge \sqrt{\frac{\lambda_\partial}{\lambda_{\min}(\mathfrak{K})\alpha_s(\frac{1}{\alpha_a}+\frac{1}{\alpha_c})}} + \frac{\lambda_g{}^2\lambda_\varepsilon{}^2(\|R^{-1}\|)^2}{4\lambda_{\min}(\mathfrak{K})}$ hold, $\dot S(t) \le 0$ will be satisfied. Finally, one can conclude that all signals are UUB.

*Case2* : Events are triggered. Consider the event-triggered controller (32) and the degree of membership update law (34) and (35).

Choosing the following Lyapunov function

$$
S_e(t) = \frac{1}{2\alpha_a}\widetilde\omega_{ae}^{\mathrm T}\widetilde\omega_{ae} + \frac{1}{2\alpha_c}\widetilde\omega_{ce}^{\mathrm T}\widetilde\omega_{ce} + \frac{\alpha_s}{\alpha_a}L_s(\xi) + \frac{\alpha_s}{\alpha_c}L_s(\xi)
\tag{46}
$$

same proof as that in *Case1*, we can demonstrate all signals are UUB.

Motivated by [14], the derivative of event-triggered function can be written as

$$
\frac{d}{dt}|\Gamma(t)| = \frac{d}{dt}(\Gamma(t)\times\Gamma(t))^{\frac{1}{2}} = \mathrm{sgn}(\Gamma(t))\dot\Gamma(t) \le |\dot u^*(\xi(t))|
\tag{47}
$$

Because all signals are UUB, absolutely existing a positive parameter $\kappa$ satisfies

$$
|\dot u^*(\xi(t))| \le \kappa
\tag{48}
$$

According to the event-triggered mechanism (28) and (29), one can derive that $\Gamma(t_d) = 0$ and $\lim_{t\to t_{d+1}}\Gamma(t_{d+1}) = \Delta|u_e^*(\xi(t))|+M$. Combining equation (47), (48) and performing some mathematical operations, the minimal inter-execution $t^* = t_{d+1} - t_d$ satisfies $t^* > \frac{|u_e^*(\xi(t))|+M}{\kappa}$, $\forall t \in [t_d,\ t_{d+1})$. Consequently, it is guaranteed that the Zeno behavior is non-occurring.

## VI. SIMULATION

In this section, YUKUN of Dalian Maritime University is utilized to verify the validity and flexibility of the optimal control strategy considering event-triggered mechanism. The parameters of YUKUN are as follows: length between perpendiculars is 105 m, beam is 18 m, rudder area is 11.46 m$^2$, loaded speed is 16.7 kn, full amidships draft is 5.2 m, full loaded displacement is 5735.5 m$^3$, block coefficient is 0.5595. Maritime environment can be set that: wind direction $\psi_{\mathrm{wind}} = 30°$, wind scale $\mathcal{S} = 6$, current direction $\psi_{\mathrm{current}} = 30°$, current velocity $v_{\mathrm{current}} = 5$kn.

Therefore, a continuous-time ship dynamic system can be considered

$$
\begin{cases}
\dot x_1 = x_2 \\
\dot x_2 = -\frac{1}{T}\left(\alpha_s x_2 + \beta_s x_2{}^3\right) + \frac{K}{T}(u+\delta_w) \\
y = x_1
\end{cases}
\tag{49}
$$

where $x_1$ and $x_2 \in \mathbb{R}$ are state variables, $u \in \mathbb{R}$ is the control input variable; reference signal $x_{1d} = \sin(\pi t/25)$; the rudder gain $K = 0.314$ and time constant $T = 62.387$; designed parameters $\alpha_s = 100$ and $\beta_s = 50$. Design parameters $\alpha_a = 0.001$, $\alpha_c = 1$, $\alpha_s = 100000$, $R = 0.067$, $\Delta = 0.39$, $M = 0.001$. The initial state can be set that $x_0 = [-0.3, 2.1, 0.1, 0.03]^{\mathrm T}$, the initial degree of membership can be set that $\omega_{a0} = [-3.4, -4, -3.5, -1.8, -2, 0, -1.4, -0.8, -1.8, -2]^{\mathrm T}$, $\omega_{c0} = [1, 1.3, 1.5, 1.3, 0, 0, 1.5, 3, 3.3, 3]^{\mathrm T}$.

Simulation results are illustrated in Fig. 1-4. The tracking trajectory and error are shown in Fig. 1, where the ship course can rapidly track the reference course in 10 seconds and tracking error can converge to a bounded compact set of zero based on the designed event-triggered adaptive optimal controller. Fig. 2 describes the general control input and the event-triggered control input. Its result illustrates event-triggered controller is superior to common controllers under the same conditions. The numerical values of event-triggered controller are smaller than that of the general controller, which effectively verifies the competent in reducing mechanical wear

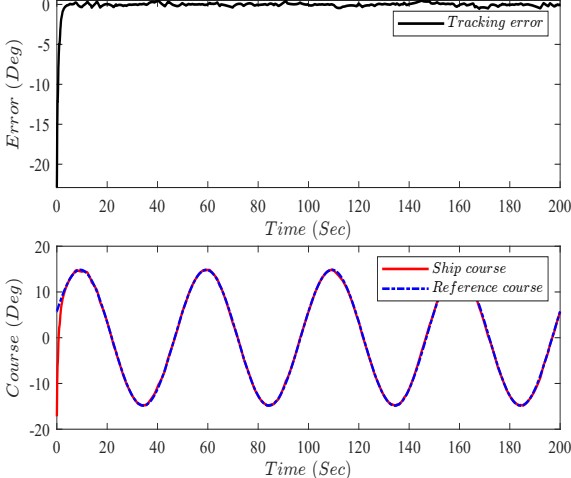

Fig. 1. Trajectories of the course tracking error, actual course and reference course.

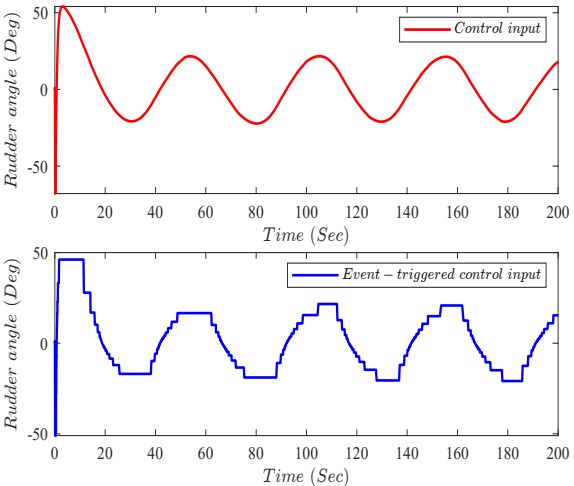

Fig. 2. Trajectories of control input and event-triggered control input.

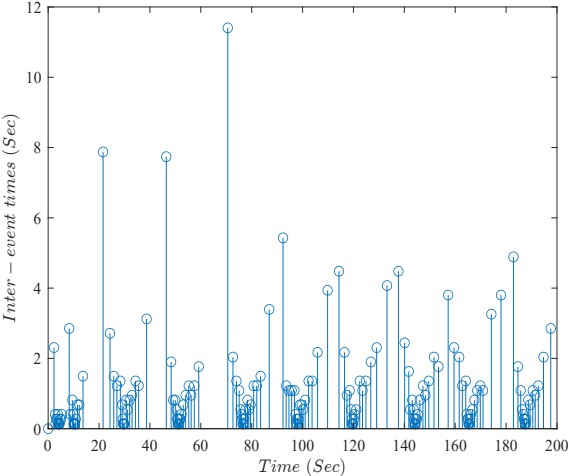

Fig. 3. Inter-event times of $u_e$.

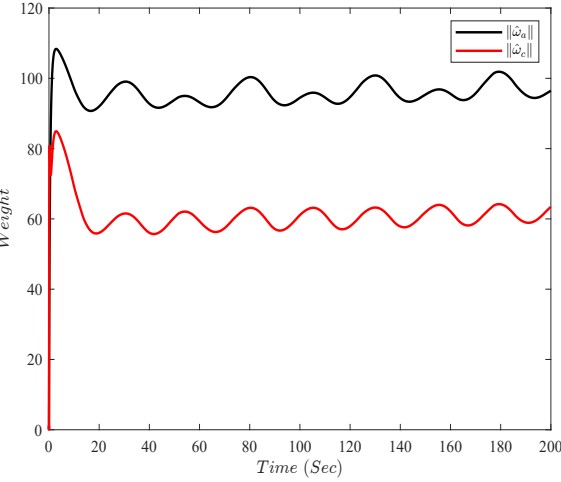

Fig. 4. Convergence situations of policy function degree of memberships $\hat{\omega}_a$ and value function degree of memberships $\hat{\omega}_c$.

and saving energy of the event-triggered mechanism. Fig. 3 describes the corresponding triggered time that highlights the advantages of cost saving for event-triggered controller. In the end, Fig. 4 gives the value function and policy function degree of memberships convergence exhibitions which demonstrate degree of membership signals can rapidly coverage to a bounded range.

## VII. CONCLUSION

In this article, an event-triggered optimal tracking control scheme has been proposed for uncertain nonlinear systems based on RL. An improved ADP technique combining actor-critic algorithm and fuzzy logic systems have been implemented in solving HJB equation of nominal system. To reduce mechanical wear of actuator and save energy, event-triggered mechanism has been performed to update controller. All signals are UUB by Lyapunov demonstration and simulations verify the feasibility of proposed scheme. In the future, we will study the tracking control problem based on deep reinforcement learning and the multi-agent systems also is an interesting direction.

### ACKNOWLEDGMENT

This work was supported in part by the Central Guidance on Local Science and Technology Development Fund of Liaoning Province (Grant No. 2023JH6/100100055); in part by the National Natural Science Foundation of China (Grant Nos. 52271360); in part by the Dalian Outstanding Young Scientific and Technological Talents Project (Grant No. 2023RY031); in part by the Basic Scientific Research Project of Liaoning Education Department (Grant No. JYTMS20230164); and in part by the Fundamental Research Funds for the Central Universities (Grant No. 3132024125).

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
