# OpenReview forum: "Event-Triggered Optimal Tracking Control for Uncertain Nonlinear System Based on Reinforcement Learning"
_IEEE.org/ICIST/2024/Conference — IEEE ICIST 2024 Conference Submission_

### Official Review · Reviewer_7FBe · 2024-08-22
**This article is very interesting and a good one**

**Rating:** 7
**Confidence:** 3

**Review:**

This paper studied an event-triggered optimal tracking control problem for uncertain nonlinear system based on RL. The obtained result is valuable and can be accepted if the following problems can be clarified.
(1) In the introduction, the shortages of those relevant studies are suggested to be further summarized.
(2) In the end of Section 1, the organization of this study is suggested to be summarized.
(3) There exist several spelling and grammar errors. Please check carefully and further polish
(4) In the Simulation, more analysis can be added to better explain the main results of this paper.
(5) The future work is missing in the Conclusion. (6) The references should be updated and their format standardized for enhanced consistency and accuracy.

---

### Official Review · Reviewer_wQXo · 2024-08-23
**Event-Triggered Optimal Tracking Control for Uncertain Nonlinear System Based on Reinforcement Learning**

**Rating:** 7
**Confidence:** 2

**Review:**

In this paper, an event-triggered optimal tracking control problem is studied for uncertain nonlinear system based on reinforcement learning. There are some problems that should be replied. Comments for this submission are given as follows:
1.The paper makes a numerous of assumptions without providing a comprehensive justification or explanation. The mathematical integrity of the paper would be enhanced by the inclusion of more detailed explanations or justifications for these assumptions.
2. Although the paper references numerous pertinent works, the literature review could be enhanced, including adaptive dynamic programming and reinforcement learning in tracking control systems.
3.How to choose the main parameter of the proposed strategy to achieve the control object ? More details should be given.
4. The main contributions are not clear enough or not expressed clearly. Please provide a more detailed explanation.

---

### Official Review · Reviewer_thrY · 2024-08-25
**Accept**

**Rating:** 7
**Confidence:** 3

**Review:**

Comment: This paper study an event-triggered optimal tracking control problem for uncertain nonlinear system based on reinforcement learning. Simulation results verify feasibility of proposed scheme. The theory is correct and can be accepted after responding the following comments.
(1) In the introduction, it is not enough to state the current work. It should be expanded and reconstructed.
(2) There are many typos and grammar errors. The authors should have a native English speaker or software packages to perform the editing check.
(3) In the simulation section, the description of the simulation results is not detailed enough.

---

### Comment · Reviewer_7FBe · 2024-08-21
**This article is very interesting and a good one**

This paper studied an event-triggered optimal tracking control problem for uncertain nonlinear system based on RL. The obtained result is valuable and can be accepted if the following problems can be clarified.
(1)	In the introduction, the shortages of those relevant studies are suggested to be further summarized.
(2)	In the end of Section 1, the organization of this study is suggested to be summarized.
(3)	There exist several spelling and grammar errors. Please check carefully and further polish
(4)	In the Simulation, more analysis can be added to better explain the main results of this paper.
(5)	The future work is missing in the Conclusion.
(6)	The references should be updated and their format standardized for enhanced consistency and accuracy.

---

### Decision · Program_Chairs · 2024-09-06

Accept (Oral)